# Characterization of a Homozygous Deletion of Steroid Hormone Biosynthesis Genes in Horse Chromosome 29 as a Risk Factor for Disorders of Sex Development and Reproduction

**DOI:** 10.3390/genes11030251

**Published:** 2020-02-27

**Authors:** Sharmila Ghosh, Brian W. Davis, Maria Rosengren, Matthew J. Jevit, Caitlin Castaneda, Carolyn Arnold, Jay Jaxheimer, Charles C. Love, Dickson D. Varner, Gabriella Lindgren, Claire M. Wade, Terje Raudsepp

**Affiliations:** 1College of Veterinary Medicine and Biomedical Sciences, Texas A&M University, College Station, TX 77843-4458, USA; sghosh@cvm.tamu.edu (S.G.); bdavis@cvm.tamu.edu (B.W.D.); mjevit@cvm.tamu.edu (M.J.J.); ccastaneda@cvm.tamu.edu (C.C.); carnold@cvm.tamu.edu (C.A.); circlezdonkeys@gmail.com (J.J.); clove@cvm.tamu.edu (C.C.L.); dvarner@cvm.tamu.edu (D.D.V.); 2Department of Animal Breeding and Genetics, Swedish University of Agricultural Sciences, 75007 Uppsala, Sweden; maria.rosengren@slu.se (M.R.); Gabriella.Lindgren@slu.se (G.L.); 3Livestock Genetics, Department of Biosystems, KU Leuven, 3001 Leuven, Belgium; 4School of Life and Environmental Sciences, The University of Sydney, NSW 2006, Australia; claire.wade@sydney.edu.au

**Keywords:** horse, disorders of sex development, cryptorchidism, reproduction, CNV, ECA29, steroid hormone, *AKR1C*

## Abstract

Disorders of sex development (DSD) and reproduction are not uncommon among horses, though knowledge about their molecular causes is sparse. Here we characterized a ~200 kb homozygous deletion in chromosome 29 at 29.7–29.9 Mb. The region contains *AKR1C* genes which function as ketosteroid reductases in steroid hormone biosynthesis, including androgens and estrogens. Mutations in *AKR1C* genes are associated with human DSDs. Deletion boundaries, sequence properties and gene content were studied by PCR and whole genome sequencing of select deletion homozygotes and control animals. Deletion analysis by PCR in 940 horses, including 622 with DSDs and reproductive problems and 318 phenotypically normal controls, detected 67 deletion homozygotes of which 79% were developmentally or reproductively abnormal. Altogether, 8–9% of all abnormal horses were homozygous for the deletion, with the highest incidence (9.4%) among cryptorchids. The deletion was found in ~4% of our phenotypically normal cohort, ~1% of global warmblood horses and ponies, and ~7% of draught breeds of general horse population as retrieved from published data. Based on the abnormal phenotype of the carriers, the functionally relevant gene content, and the low incidence in general population, we consider the deletion in chromosome 29 as a risk factor for equine DSDs and reproductive disorders.

## 1. Introduction

Mammalian disorders of sex development (DSDs) encompass a broad range of complex congenital conditions that affect sex determination, sexual differentiation, and the development and function of gonads [1,2,3,4,5,6]. The majority of DSDs are characterized by an atypical combination of chromosomal, gonadal and phenotypic sex and are broadly classified as XX and XY DSDs [5,7]. Though, in some forms of XY DSDs, such as cryptorchidism and hypospadias, the genetic and gonadal/phenotypic sex are in concordance [8]. The prevalence of DSDs in humans is approximately 1.7% of all live births [7,9], whereas the frequency of non-syndromic cryptorchidism alone can be as high as 8% among full-term male births depending on the geographical area [8,10].

Sex development is a complex multi-step process controlled by many genes acting in networks [4,5,11], which complicates the search for the molecular causes of DSDs. Only a handful of human and mouse DSDs can be attributed to point mutations or copy number variants (CNVs) in a single gene, others are thought to be oligogenic with the involvement of additional modifier genes, while molecular causes of many conditions are not known. Furthermore, variations in the same gene can result in different DSD phenotypes and similar DSD phenotypes may have different genetic causes [4,11]. 

Multiple forms of XX and XY DSDs have also been described in horses [12,13,14,15] and all forms affect equine health and reproduction, as well as the economy of breeders and owners. Among equine XY DSDs, the most common is cryptorchidism with a prevalence of 2–8% among male births [16], followed by a spectrum of XY and XX DSDs, all characterized by various degrees of discrepancy between genetic, gonadal and phenotypic sex and ambiguous external sex phenotypes [12,13,14,17]. 

Knowledge about the molecular causes of equine DSDs is sparse. For example, no genes or mutations are known for any of the XX DSDs with intersex, hermaphrodite, pseudohermaphrodite or ambiguous sex phenotypes [12,13,14]. Among XY DSDs, the *SRY*-negative forms are associated with various Y chromosome deletions [18], while genetic causes of *SRY*-positive XY DSDs are known only for a few cases and are heterogeneous. In three families of different breeds, *SRY*-positive XY DSDs have been associated with different mutations in the androgen receptor gene resulting in androgen insensitivity syndrome [19,20,21]. Earlier studies identified a large deletion in horse (*Equus caballus*, ECA) chromosome 29 (ECA29) that was found in two related Standardbred horses with *SRY*-positive XY DSDs and male pseudohermaphrodite phenotypes [22]. The deletion involves steroid hormone biosynthesis genes belonging to the aldo-keto reductase family 1 C (*AKR1C*), of which mutations have been associated with XY DSDs, such as cryptorchidism, undervirilized external genitalia, dihydrotestosterone (DHT) deficiency and sex reversal conditions in humans [1,3,4].

In this study, we will further investigate the molecular landscape of the deletion in horse chromosome 29 by PCR, fluorescence in situ hybridization (FISH) and whole genome sequencing (WGS). We characterize the deletion breakpoints, develop a simple PCR test for the detection of deletion homozygotes, and determine their frequency in general equine populations, as well as among horses with various DSDs and reproductive disorders.

## 2. Materials and Methods 

### 2.1. Ethics Statement

Procurement of blood and hair samples for DNA isolation followed the United States Government Principles for the Utilization and Care of Vertebrate Animals Used in Testing, Research and Training. These protocols were approved as AUPs #2012-0250 and #2018-0342 CA at Texas A&M University. Owners of the Swedish samples gave permission for their animals to be used in the study and the study was approved by the Ethics Committee for Animal Experiments in Uppsala, Sweden (#C 121/14 and #5.8.18–15453/2017).

### 2.2. Horses and Phenotypes

Samples of 940 horses of over 45 diverse breeds and breed mixes were used for experiments in this study (Appendix A). These included 364 horses with various DSDs of which 159 were cryptorchids; 258 phenotypically normal but reproductively or hormonally abnormal male and female horses, and 318 control horses with normal male or female phenotypes, but with limited information about fertility. Control horses included Thoroughbred mare *Twilight*, the DNA donor for horse reference genome assemblies EquCab2 [23] and EquCab3 [24], and her half-sibling Thoroughbred stallion *Bravo*, the DNA donor of bacterial artificial chromosome (BAC) library CHORI 241 (https://bacpacresources.org/). Both horses are normal with proven fertility. The majority of samples available from the depository of Texas A&M Molecular Cytogenetics laboratory also had karyotype data. 

In addition, whole genome sequence (WGS) and/or high density 670K SNP genotyping data from publications [25,26,27] were used for population analysis. Horses from these cohorts were not specifically phenotyped for sex development or fertility.

### 2.3. Genomic and BAC DNA Isolation and Quality Control

Genomic DNA was isolated from hair or peripheral blood using Gentra Puregene Blood Kit (Qiagen) following the manufacturer’s protocol. The DNA was checked for quality and quantity with Nanodrop 2000 spectrophotometer (Thermo Scientific). 

The IDs and sequence coordinates of 4 partially overlapping genomic clones spanning the deletion in ECA29 and a control clone outside the deletion (Table 1) were retrieved from NCBI (https://www.ncbi.nlm.nih.gov/genome/?term=horse). The clones were picked from the CHORI 241 BAC library (https://bacpacresources.org/). High molecular weight BAC DNA was isolated with High Pure Plasmid Isolation Kit (Roche Diagnostics) per manufacturer instructions and checked for molecular weight, quality and quantity with Agilent 2100 TapeStation system. 

### 2.4. Primers and PCR Analysis

The ECA29 deletion was analyzed by qualitative and quantitative PCR. Primers for sequence tagged sites (STSs) around putative deletion breakpoints and for genes inside the deletion (Appendix A) were designed using horse reference genome EquCab3 [24] (NCBI: https://www.ncbi.nlm.nih.gov/genome/?term=horse) and Primer3 software [28]. Each primer was validated by in silico PCR in UCSC (http://genome.ucsc.edu/) to produce a single product unique for ECA29. 

Qualitative PCR was performed with JumpStart Taq DNA polymerase (Sigma) and the results were analyzed by agarose gel electrophoresis. Qualitative PCR with DELin1 primers to detect horses with homozygous deletion were carried out in duplicate reactions and duplicate experiments.

Quantitative PCR (qPCR) experiments were performed with LightCycler® 480 (Roche Diagnostics) in triplicate assays and triplicate 20 µL reactions, each containing 50 ng of template DNA, 10 μM primers, and the Hot FirePol® EvaGreen® qPCR Mix Plus (Solis BioDyne). Relative copy numbers were determined in comparison to the reference sample (Thoroughbred *Bravo*) and normalized to an autosomal reference gene *GAPDH*. Copy numbers were quantified by the comparative C_T_ method (2^ΔΔ^Ct) [29,30] with *p* < 0.05 as a cut-off threshold for statistical significance. 

### 2.5. Fluorescence In Situ Hybridization (FISH)

The DNA (1 µg) of BACs 23N13 spanning the deletion and a control BAC 76H13 outside the deletion (Table 1) were labeled with biotin-16-dUTP or digoxigenin-11-dUTP using Biotin- or DIG-Nick Translation Mix (Roche Diagnostics). Labeled probes were hybridized to metaphase chromosomes of horses where analysis by PCR suggested homozygous or heterozygous deletion, and to control horses without deletion. FISH was done following standard protocols [31]. Images for a minimum of 20 metaphase cells were captured for each experiment and analyzed with a Zeiss Axioplan2 fluorescent microscope equipped with Isis v5.2 (MetaSystems GmbH) software.

### 2.6. Sequencing and Bioinformatics Analysis

Whole genomes of two horses with homozygous deletion in ECA29 and XY DSD phenotypes (H369, a male pseudohermaphrodite and H354, a bilateral cryptorchid; Appendix A) were sequenced using 1x100 bp libraries prepared with TruSeq DNA PCR-Free Sample Preparation Kit (Illumina), and an Illumina HiSeq2500 platform producing approximately 6-10X average coverage per animal. All data were aligned to EquCab3 using bwa [32], sorted with samtools 1.9 [33], and duplicates marked with sambamba [34]. Publicly available control data from Jagannathan et al. 2019 (QH070 and HN001 for visualization) were mapped in the same fashion.

The 4 BAC clones (Table 1) spanning the deletion were sequenced by Pacific Biosciences (PacBio) single molecule real time (SMRT) technology. Sequencing libraries with a molecule size range of 20 kb were generated using the RS DNA Template Preparation Kit (Pacific Biosciences), loaded on a SMRT cell and sequenced for long-reads with C2 chemistry on the PacBio RS [35], with an estimated yield of approximately 5 Gb data per SMRT cell. Resulting filtered subreads were assembled using canu 1.8 [36] for each individual BAC, as well as a separate assembly for all data. For reference based alignment, PacBio subreads were aligned to EquCab3 using minimap2.1 [37]. Furthermore, mapping of individual contigs generated by canu were aligned using minimap for visualization.

The content and functions of genes affected by the deletion were retrieved from NCBI (http://genome.ucsc.edu/), UCSC (http://genome.ucsc.edu/) and Ensembl (http://www.ensembl.org/index.html) Genome Browsers, and GeneCards (http://www.genecards.org/).

Protein sequences for horse, human and mouse *AKR1C* genes were retrieved from NCBI (http://genome.ucsc.edu/) and aligned with the Clustal Omega multiple sequence alignment tool (https://www.ebi.ac.uk/Tools/msa/clustalo/).

## 3. Results

### 3.1. Identification of Additional Deletion Carriers

The initial discovery of the deletion in ECA29 was by array comparative genomic hybridization (aCGH), which identified a ~200 kb homozygous deletion in two related horses (H348 and H369; Appendix A) with XY DSDs [22]. To determine, whether or not the deletion is an isolated event we used the published [22] deletion-specific primers (*DelIn1*, Appendix A) to screen a small selection of horses with XY DSDs by regular PCR. As a result, additional animals that were homozygous for the deleted sequence were identified (Figure 1), indicating that the deletion is not limited to the two discovery horses and requires more detailed analysis.

### 3.2. Demarcation Deletion Breakpoints

In order to demarcate deletion breakpoints, a set of primers were designed for PCR walking around the putative deletion start (primers S1-S11) and end sites (primers E1–E9) (Appendix A, Figure 2), which were determined based on aCGH data [22]. Walking by PCR using DNA from horses with known homozygous deletion (D/D) and controls without deletion suggested that the deletion starts in a 1019 bp region at chr29: 29,740,911-29,741,930 between primers S7 and S8, and ends in a 95 bp region at chr29:29,944,184-29,944,279 between primers E8 and E9 (Appendix A, Figure 2). However, demarcation of the end of the deletion by PCR remained dubious because sequences corresponding to the amplicons of primers E10–E12 showed multiple hits in EquCab3. Furthermore, primers E4 did not amplify from D/D or controls (Appendix A), suggesting that the region is complex and the reference assembly has gaps and/or errors in this region.

To determine the deletion boundaries more accurately, we generated WGS for two D/D horses: H354 and H369 (Figure 1, Appendix A). Sequence alignment with EquCab3 and published WGSs of 2 normal horses [27] showed that the deletion in ECA29 starts at position 29,741,143 bp in H354 and at 29,741,147 bp in H369, near a documented repetitive region in EquCab3, and consistent with the region determined by PCR (Figure 3). In horse H354, the putative end was around 29,944,505 bp and in horse H369, at 29,952,177 bp, thus over 7.6 kb more distal. Both putative deletion end sites determined by WGS (Figure 3) extended further compared to the one determined by PCR (Figure 2). Based on WGS data, we estimated the approximate size of the deletion to be 200–210 kb. However, due to multi-mapping short read alignment from 29,944,163 to 29,952,267 roughly 20–30X average depth, the precise end breakpoint was not precisely determined. This ~8,100 bp region indicates a putative tandem duplication or complex rearrangement at the end of the deletion, potentially inducing the structural change. No duplication is evident in 88 control horses from the WGS data obtained from Jagannathan et al. 2019, thus genotypically normal horses do not possess the putative additional structural variant(s) associated with the deletion. Resolution is unattainable with short-read data, and should be explored using newer PacBio or Oxford Nanopore long-read data. 

In an attempt to improve sequence assembly in this region, we de novo assembled long-read PacBio sequence data of 4 partially overlapping BACs spanning the deletion (Table 1, Figure 3). Previously, we have shown that Thoroughbreds *Bravo* and *Twilight*, the DNA donors of the BAC library and genome assembly, respectively, differ by just 3 CNVs across the genome with no CNVs in ECA29 [22]. Thus, the BAC sequences should closely resemble the reference genome and can be used to improve the assembly in this region. Independent assembly of all four BACs produced one long contig with high read coverage (>3000) each. Assembly of all data together produced one 338,300 bp contig spanning the deletion. This confirmed that there is no evidence for duplication in the region, other than in D/D individuals (Figure 3).

Taken together, both PCR analysis and WGS confirmed the presence of a large, approximately 200 kb deletion in some horses with XY DSDs. While precise demarcation of deletion boundaries remained tentative due to sequence complexity in the region, the most proximal and most distal points of the deletion by WGS were chr29:29,741,143–29,952,177 spanning 211,034 bp.

### 3.3. Gene Content of the Deleted Region

The deleted region chr29:29,741,143–29,952,177 contains 2 predicted protein-coding genes and one RNA gene by Ensembl (Figure 3). However, the Ensembl gene prediction ENSECAG00000021292 corresponds to 5 non-overlapping fragments of the same gene in NCBI (Table 2), suggesting that the region contains members of a gene family which have not yet been accurately identified and annotated. Despite this, all predicted coding genes in the deleted region belong to aldo-keto reductase family 1, known to play essential roles in the metabolism of all steroid hormones [38], including the production of extra-testicular androgens [39]. Deletion of all genes in this region in select XY DSD horses with homozygous deletion was confirmed by PCR with gene-specific primers E661, E668, E458, E332 and ED2 (Figure 3; Appendix A).

### 3.4. Detection of Deletion Heterozygotes by Quantitative PCR and FISH

Once approximate deletion start and end sites had been determined (Figure 2 and Figure 3), conventional qualitative PCR with primers designed inside the deletion (Appendix A) was a fast and efficient approach for the detection of horses homozygous for the deletion (Figure 1 and Figure 2). However, because of the large size (~200 kb) of the deleted region and ambiguity in defining deletion boundaries, we were not able to detect heterozygous carriers by qualitative PCR. Therefore, we designed gene-specific primers G_2879 (Figure 3, Appendix A) for quantitative PCR (qPCR) and analyzed selected 12 horses with DSDs (including 8 cryptorchids) and 11 normal control horses for possible heterozygotes (Figure 4). As expected, previously known D/D horses had zero copies. Other horses, both with DSDs and normal controls, showed considerable CNV in a range from 20 to 120 relative copies, which is consistent with previous CNV studies in the region [22]. 

The broad range of variation in qPCR results made calling deletion heterozygotes ambiguous and we further validated the results by FISH. We selected two horses with relative copy numbers below 50 (*Bravo*—normal fertile male and H657—an XX intersex), one horse with > 50 copies (H395), and one with homozygous deletion and zero copies (H369) as a negative control. FISH was done with BAC 23N13, which spans the deletion (Figure 3) and a control BAC 76H13, which maps to the proximal part of ECA29 (Table 1). The results showed that H395 (>50 copies) and *Bravo* (<50 copies) did not have the deletion, horse H657 (<50 copies) with similar copy numbers by qPCR as *Bravo* was heterozygous, and horse H369, as expected, homozygous for the deletion (Figure 5). These results indicate that FISH analysis can clearly discriminate between horses with no deletion, deletion heterozygotes and horses with homozygous deletion. However, FISH is a method of too low throughput for routine screening of large number of horses. On the other hand, relative copy number estimation by qPCR to discriminate between horses with no deletion and those with heterozygous deletion remained ambiguous. Furthermore, a broad range in copy numbers as detected by qPCR analysis (Figure 4) suggests that this CNV may be multi-allelic like several common complex CNVs described in human diseases [40]. This also implies that qPCR-based analysis can confidently detect zero copies in homozygous deletion, maybe also 1 copy in simple heterozygotes, but will provide no information about the distribution of >2 copies on homologous chromosomes. 

### 3.5. Frequency of the Deletion Among Horses with Reproductive Problems and DSDs, and in General Population

Because the first identified homozygotes for ECA29 deletion were horses with DSDs [22] (Figure 1), and because the deletion involved genes necessary for steroid hormone biosynthesis (Table 2) and sex development [38], we further investigated by PCR the frequency of deletion homozygotes among 622 developmentally or reproductively abnormal horses. These included horses with various XX and XY DSDs and other congenital disorders, horses with normal male or female sex development but with unexplained infertility or subfertility, and horses producing offspring with DSDs. We also tested 318 phenotypically normal male and female horses, though with limited information about their fertility (63 with proven fertility; 255 with no information on fertility). Details about all 940 individual horses are presented in Appendix A. 

Altogether, we identified 67 horses with homozygous deletion of which 53 horses (79%) were diagnosed with DSDs or fertility problems. The frequency of homozygous deletion was almost two times higher in the developmentally and reproductively abnormal cohort (8.1%) compared to controls (4.7%), with the highest incidence found in cryptorchid horses (9.4%) (Table 3). We did not observe any bias of the deletion between bi- or unilateral cryptorchids. There was also no difference between the occurrence of the deletion across different breeds and D/D individuals were present in 17 different breeds and breed mixes out of 45 breeds involved.

In order to estimate the frequency of the homozygous deletion in the general equine population, we analyzed publicly available WGS and 670K SNP array data for the region chr29: 29,741,143—29,952,177 (Table 4). The frequency of homozygous deletion carriers varies considerably between the studies being 0–1% among global warmblood and pony breeds [26,27], but reaching over 7% in a recent study, which is strongly biased towards draught horses [25]. Collectively, the frequency of homozygous deletion in general equine population is lower (5.6%) than that among horses with DSDs and fertility problems (8.1%). Because the sequence assembly of the region has ambiguities (Figure 3), and the region likely represents a complex multi-allelic CNV, no efforts were made to estimate the number of putative deletion heterozygotes.

## 4. Discussion

Here we characterized a ~200 kb CNV region (CNVR) in horse chromosome 29 at 29.7–29.9 Mb, which contains genes essential for steroid hormone biosynthesis and was deleted from both homologs in 8–9% of horses with reproductive problems and DSDs. On the other hand, 79% (53/67) of the horses carrying the homozygous deletion were developmentally or reproductively abnormal.

The deleted region in ECA29 is embedded in a larger common CNVR that has been found in the general equine population of diverse origins by several previous studies [22,25,41,42]. The most extensive recent genome-wide analysis of 939 CNVRs in 1755 horses of 8 European breeds, detected the larger 714 kb common deletion-duplication CNVR (CNVR915) in 1204 (68.6%) horses [25]. As a notable contrast, the homozygous deletion of the smaller 200 kb region is rare and was found only in about 4.7% of phenotypically normal horses used in this study, and in 0% to 7.6% of general equine population from other studies (Table 3 and Table 4). Based on published data, the frequency of the homozygous deletion is extremely low (0% to 1%) among 26 global warmblood and pony breeds [26,27]. For example, the larger CNVR is present in 84% (216/256) of Exmoor ponies studied for CNVs [25], but none of these carry the homozygous deletion of the smaller region [26] (Table 4). At the same time, the frequency of D/D individuals seems to be considerably higher (7.6%) among European draught breeds [25] and it would be of future interest to compare the genotyping data with reproductive phenotypes in these breeds. 

Given that 79% of the carriers of the homozygous deletion in ECA29, as determined in this study, were developmentally or reproductively abnormal, the content of genes in the region is of particular interest. According to gene predictions, both deleted protein-coding genes belong to the aldo-keto reductase gene family 1C (*AKR1C*) (Figure 3, Table 2) members of which carry out various functions in steroid hormone metabolism. More specifically, AKR1C enzymes (AKR1C1–AKR1C4) function in vivo as 3-keto-, 17-keto- and 20-ketosteroid reductases and metabolize a broad spectrum of natural and synthetic therapeutic steroids [6]. They are involved both in androgen and estrogen metabolism by controlling concentrations of active androgens and estrogens, and regulating occupancy and trans-activation of androgen and estrogen receptors [38]. All pathways to the potent androgens testosterone and dihydrotestosterone (DHT) proceed through family members *AKR1C2-AKR1C4* [38], while *AKR1C2* and *AKR1C4* also participate in ‘the backdoor pathway’ that leads to DHT synthesis without testosterone intermediate [43,44,45]. 

Altered expression of individual *AKR1C* genes is related to the development of prostate and breast cancer, and endometriosis in humans due to changes in progesterone, estrogen and androgen metabolism [38,46]. Mutations in *AKR1C* genes are associated with human XY DSDs, including cryptorchidism, sex reversal and defective androgen synthesis phenotypes [3,4,38,44,47]. 

Mouse models are less informative because there are no one-to-one mouse orthologs for human or horse *AKR1C* genes (Ensembl: http://www.ensembl.org/index.html). While amino acid sequence identity between human *AKR1C1-AKR1C4* paralogs is 82–99% (Appendix A) [38], their one-to-one orthologs have been found only in primates, while orthologs in other mammals, including mouse, are many-to-many type (Ensembl: http://www.ensembl.org/index.html). On the one hand, this suggests rapid evolutionary divergence of *AKR1C* orthologs across species, and on the other hand, difficulties to annotate paralogs within a species. The closest murine homolog for human *AKR1C1-AKR1C4* genes is *Akr1c18* (MGI: http://www.informatics.jax.org/marker) sharing only 65–68% amino acid sequence identity with human and 51–65% with equine orthologs (Appendix A). Nevertheless, it is noteworthy that *Akr1c18* (-/-) live knockouts are exclusively female with reduced fetal viability, low adult pregnancy rates and increased embryonic loss, whereas male knockouts are likely not viable [48,49]. 

The human and mouse data strongly suggest that *AKR1C* genes are necessary for normal sex development and reproduction and their mutations have consequences. Furthermore, in the CNV map of the human genome, the *AKR1C* genes are not among the ~100 “expendable” CNV genes which can be homozygously deleted with no deleterious effect [50]. This is consistent with the findings of the present study where the majority (79%) of horses with the homozygous deletion were abnormal regarding sex development and/or reproduction. Though, there is no single clear explanation for the remaining 21% of horses also carrying the homozygous deletion, in part because the available fertility data for the “normal” population were sparse (Appendix A). Furthermore, the assembly of the region in ECA29 is incomplete even in the recently improved reference genome EquCab3 [24], and annotation of individual *AKR1C* family members is ambiguous (Table 2). The region is structurally complex, containing CNVs, segmental duplications and *AKR1C* paralogs that share over 86% sequence identity with each other [6,51]. These features complicated accurate demarcation of deletion boundaries. For example, by combining PCR and WGS data, we confined the deletion between 29.7–29.9 Mb (Figure 2 and Figure 3). In EquCab3 in UCSC Genome Browser (https://genome.ucsc.edu/), the cluster of *AKR1C* paralogs is located at chr29: 29.7–30.3 Mb, thus some members of the gene family may escape the deletion and compensate the loss of others. It is also possible that the size of the deletion differs between individual horses, but was not captured by PCR analysis of a short target region. However, perhaps the most plausible explanation for why not all animals with the deletion showed abnormal phenotypes, as well as why those with the deletion had a broad spectrum of abnormal phenotypes, stays in the very nature of the genetic regulation of steroid metabolism. It is well documented that among all CNV genes in humans and animals, the most abundant are G-protein coupled receptors, and immunity and steroid metabolism genes [22,50,52,53]. This implies that the gene networks and biochemical pathways governed by these genes are so critically important for the development, viability and adaptive plasticity that they are duplicated and backed up by alternatives in the genome [54]. We therefore hypothesize that phenotypic consequences of the deletion in ECA29 varies from case to case depending on the overall genetic background of each individual.

Further, in the large cohort of 622 developmentally and reproductively abnormal horses, only 8%-9% carried the homozygous deletion and phenotypes of the carriers were diverse. Phenotypically, the most uniform group were cryptorchids of which 9.4% were homozygous for the deletion (Table 3), though cryptorchid phenotype too varied in a broad range. Therefore, it was not possible to statistically associate the deletion with any specific form of equine cryptorchidism or other DSD. Our observations are consistent with those in humans where studies on DSDs and disorders of reproduction are more advanced and abundant [1,2,3,4], but face similar challenges for uncovering the genetic causes. One is the very nature of these conditions which prevents reproduction and transmission, making almost every case a singleton. Another is the complex gene network underlying sex development and reproduction. In addition, most of these conditions have a spectrum of phenotypes. Together these factors exclude association studies based on large, well-phenotyped case-control cohorts, and require genetic studies on a case-by-case basis. 

Nevertheless, based on the size and gene content of the deletion, the abnormal phenotype of the majority (79%) of deletion homozygotes, and the low frequency (~5%) of the homozygous deletion in general equine population, we consider this deletion as a possible causative, but definitely a risk factor, for some forms of equine DSDs, cryptorchidism, and reproductive disorders.

## 5. Conclusions and Future Approaches

We characterized a homozygous ~200 kb deletion in ECA29 as a risk factor for equine DSDs and reproductive disorders. These conclusions were based on the following notions: i) the deletion eliminates functionally relevant *AKR1C* genes, which have been associated with DSDs in humans; ii) the frequency of the deletion was higher in abnormal horses (8–9%) compared to normal population (1–5%), and iii) the majority (79%) of horses with the homozygous deletion were phenotypically abnormal. Even though denoted as a *risk factor* because of difficulties to establish statistically sound causative genotype–phenotype relationship, the findings add to the currently very short list of genomic regions underlying equine disorders of sex development and reproduction (reviewed by [55]). A simple PCR analysis to identify ECA29 deletion homozygotes can be easily included in clinical genetic testing of problem horses, similarly to the routinely conducted test for the Y-linked *SRY* gene [18]. At the same time, for any further progress in the study of this deletion, it is important to improve phenotype descriptions for all new cases. For example, since the *AKR1C* genes are involved in steroid hormone biosynthesis, it would be important to include data on serum hormone levels. Further, the fact that only 8–9% of the 622 reproductively and developmentally abnormal horses in this study carried the ECA29 deletion also implies the need of continuing efforts to decipher the molecular causes of equine DSDs and reproductive disorders. High expectations are put on personalized genomics by generating WGS data for large numbers of individual horses, as well as on the ongoing equine initiative of the Functional Annotation of Animal Genomes (FAANG) project (reviewed by [55]) to identify important functional elements in the horse genome, particularly those underlying complex traits such as development and reproduction.

## Figures and Tables

**Figure 1 genes-11-00251-f001:**
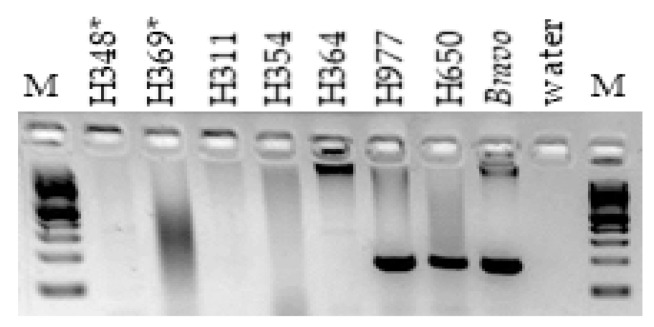
Discovery of homozygous deletion in horses with XY DSDs using PCR and *DelIn1* primers; horses H348 and H369 denoted with an asterisk are the two related male pseudohermaphrodites where the deletion was initially discovered by aCGH; horses H311, H354 and H364 are cryptorchids; H977, H650 and *Bravo* are normal controls; M—100 bp ladder (NEB).

**Figure 2 genes-11-00251-f002:**
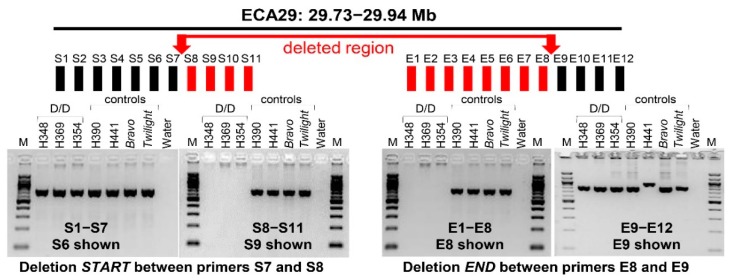
Demarcation of deletion breakpoints by PCR walking. Black horizontal bar at the top denotes the span of PCR walking in ECA29 between 29.73 and 29.94 Mb. Rectangles S1–S11 and E1–E12 denote sequence tagged sites (STSs) around the deletion start and end, respectively. STSs shown with black rectangles were located outside the deletion and STSs with red rectangles were inside the deletion. Red arrows indicate the deletion start and end sites. Representative agarose gel images with PCR results using STS primers outside and inside the deletion and gDNA from horses with homozygous deletion (H348, H369, H354) and 4 control horses are shown below the two deletion breakpoints. Details about PCR walking primers are in Appendix A.

**Figure 3 genes-11-00251-f003:**
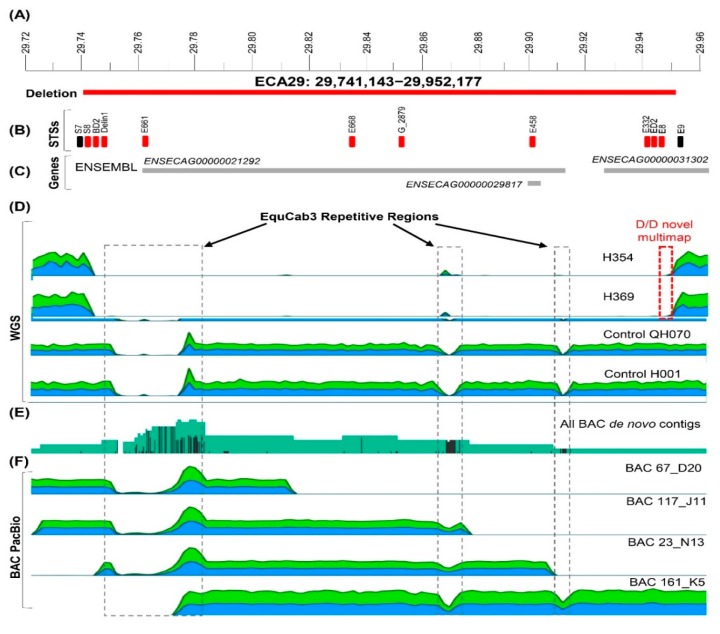
Schematic overview of the deleted region in ECA29: 29,741,143–29,952,177 (the approximate maximum span of the deletion based on whole genome sequence (WGS))**. (A)** Position of the deletion in ECA29 sequence map; **(B)** STSs in the region that were used for breakpoint analysis, detection of deletion homozygotes and to confirm homozygous deletion of specific genes; (**C)** Gene map of the region according to NCBI and Ensembl; **(D)** Relative read coverage plots for WGS of two cases and two control horses delineating the deleted region. Boxed in grey are known repetitive regions in EquCab3. Boxed in red is the structural variant present only with the documented deletion in cases, and (**E)** All de novo contigs assembled from BACs spanning the region of the deletion; (**F)** Relative PacBio read coverage plots when aligned to EquCab3. In all sequence coverage plots, green color indicates plus-strand and blue color minus-strand alignment.

**Figure 4 genes-11-00251-f004:**
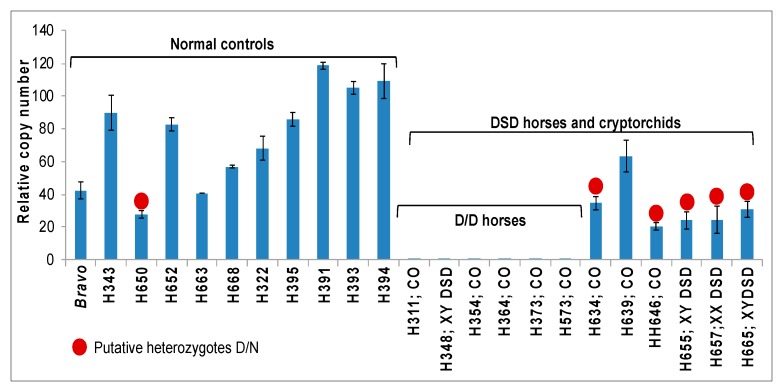
Quantitative PCR for the detection of putative deletion heterozygotes. Y-axis: relative copy number; x-axis: individual horses tested, including 11 normal controls at the left and 12 horses with DSDs at the right; CO—cryptorchid; red dots denote putative heterozygotes based on relative copy numbers.

**Figure 5 genes-11-00251-f005:**
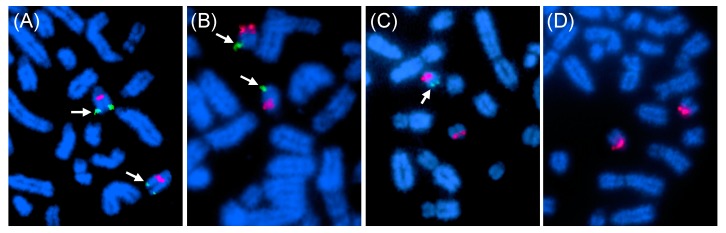
Deletion analysis by FISH with deletion-specific BAC 23N13 (green; arrows) and a control BAC 76H13 (red). **(A)** Normal male control *Bravo* with <50 copies by qPCR and no deletion by FISH; **(B)** Normal male H395 with >50 copies by qPCR and no deletion by FISH; **(C)** XX DSD horse H657 with <50 copies by qPCR and heterozygous deletion by FISH; and **(D)** XY DSD horse H369 with homozygous deletion by qPCR and FISH.

**Table 1 genes-11-00251-t001:** Information about ECA29 bacterial artificial chromosome (BAC) clones used for sequencing and fluorescence in situ hybridization (FISH) analysis.

CHORI-241 BAC	Insert Size, bp	Location in EquCab3	Use in This Study
076H13	185,817	4,148,266-4,334,082	FISH, control
067D20	136,746	29,675,975-29,812,720	PacBio
177J11	151,587	29,723,325-29,874,911	PacBio
023N13	159,470	29,745,215-29,904,684	PacBio and FISH
161K5	223,786	29,771,429-29,995,214	PacBio

**Table 2 genes-11-00251-t002:** Predicted genes in the deleted region by Ensembl and their counterparts by NCBI annotation.

Ensembl	Gene Symbol	Gene Name	Ensembl Transcripts	Ensembl Coordinates	NCBI	NCBI Transcripts	NCBI Coordinates
ENSECAG 00000021292	*AKR1C3*	Prostaglandin F synthase 1; Aldo-Keto Reductase Family 1 Member C3	10	29,762,757–29,916,689	LOC 100070491	1; partial mRNA; low quality protein	29,763,330–29,777,989
					LOC 100070501	3	29,783,899–29,797,702
					LOC 100070509	3	29,821,260–29,834,688
					LOC 100070516	1	29,840,106–29,854,539
					LOC 100057212	2	29,890,576–29,916,671
ENSECAG 00000031302	*AKR1C23L*	Aldo-keto reductase family 1 member C23-like protein	7	29,928,015–30,063,271	LOC 100070528	6	29,928,409–29,944,738
ENSECAG 00000029817	U6	U6 splicesomal RNA gene	1	29,911,416–29,911,522	LOC 111771275	1	29,911,416–29,911,522

**Table 3 genes-11-00251-t003:** Frequency of deletion homozygotes among horses with DSDs and/or fertility problems.

Phenotype	No. of Horses	Homozygotes, DD	Percent
DSD (non-cryptorchid)	205	17	8.3
Cryptorchid	159	15	9.4
Normal but subfertile/infertile	258	21	8.1
All DSDs, subfertile, infertile	622	53	8.1
Normal controls	318	14	4.7

**Table 4 genes-11-00251-t004:** Occurrence of ECA29 deletion in general equine population.

No. of Horses	Method	No. of Breeds	Breeds	D/D	D/D, %	Reference
280	670K	1	Exmoor pony	0	0	[26]
88	WGS	25	Akhal-Teke, American Paint, American Standardbred, Arabian, Polish Warmblood, Badenwürttemberg Warmblood, Bayer Warmblood, German Riding Pony; Holsteiner, Morgan, Franches-Montagnes, Hannoverian, Haflinger, Icelandic, Dutch Warmblood, Noriker, Oldenburger, American Quarter Horse, Swiss Warmblood, Shetland Pony, Thoroughbred, Trakehner, UK Warmblood, Westfale, Welsh Pony	1	1.1	[27]
1755	670K	8	Ardenner, Belgian Draught, German Draught, Exmoor Pony, Vlaams Paard, Belgian Warmblood, Swedish Warmblood, Friesian	135	7.6	[25]
**TOTAL: 2403**		**32**		**136**	**5.6**

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
