# Peer review of "Characterization of a Homozygous Deletion of Steroid Hormone Biosynthesis Genes in Horse Chromosome 29 as a Risk Factor for Disorders of Sex Development and Reproduction"

_genes, 2020, doi:10.3390/genes11030251_

Round 1

Reviewer 1 Report

In this study, Ghosh et al., characterized a ~200 kb deletion in the horse chromosome ECA29 (at 29.7 - 29.9 Mb) by using PCR analysis and whole genome sequencing analysis in 940 horses, including 622 horses with disorders of sex development and reproductive problems and 318 horses with normal phenotype.

Major comment

The study mainly emphasizes that the deletion region in horses contain AKR1C genes, which functionally involved in the steroid hormone biosynthesis, including androgens and estrogens, as reported in humans. However, the identification and annotation of this gene family in horse is not well-understood and could limit the interest of this study. Figure 3 and table 2 shows a list of only two members (AKR1C3 and AKR1C23L) with multiple predicted gene IDs. Reports solely based on these predicted information is risky because this information might be deleted or moved between chromosomes/genes when the next version of horse genome is released by the Ensembl/NCBI. Therefore, more analysis such as RACE PCR or extensive bioinformatics analysis on ~3000 – 5000 bp (including the deletion region) for the search of potential AKR1C promoter region, open reading frame, and protein codes in horse compared with human is necessary.    

Minor comments

Please provide more detailed information in the figure legends 2, 4, and 5.

Author Response

Response to Reviewers

We thank the reviewers for the thorough, insightful, and constructive critique. As follows, please find our responses and list of revisions made in the manuscript. Please let us know if you would like us to provide any further information or clarification.

REVIEWER 1

 Major comment: The study mainly emphasizes that the deletion region in horses contain AKR1C genes, which functionally involved in the steroid hormone biosynthesis, including androgens and estrogens, as reported in humans. However, the identification and annotation of this gene family in horse is not well-understood and could limit the interest of this study. Figure 3 and table 2 shows a list of only two members (AKR1C3 and AKR1C23L) with multiple predicted gene IDs. Reports solely based on these predicted information is risky because this information might be deleted or moved between chromosomes/genes when the next version of horse genome is released by the Ensembl/NCBI. Therefore, more analysis such as RACE PCR or extensive bioinformatics analysis on ~3000 – 5000 bp (including the deletion region) for the search of potential AKR1C promoter region, open reading frame, and protein codes in horse compared with human is necessary.

 Response: We agree that horse genome annotation in this region is incomplete, which we have underscored in the text (3.3. Gene content of the deleted region) and in Table 2 by presenting all NCBI and Ensembl annotations for this region. However, we respectfully disagree that using this predicted information is risky and requires further analysis for ORFs and promoters. Our argument is based on the fact that both in EquCab2 and EquCab3 all members of AKR1C gene family have been assigned to a defined region in chr29, and have been annotated as protein coding. The latter infers that the annotation pipelines have recognized ORFs and promoters. Likewise, all human and mouse AKR1C family genes are identified as protein coding and map to defined regions in HSA10p and MMU13, respectively. Both human and mouse chromosomes share evolutionarily conserved synteny with horse chr29. Thus, while we expect that the ongoing functional annotation initiative of the horse genome (equine FAANG; Note that for RACE PCR we need a collection of RNA-preserved tissues from these horses and we do not have those) will refine gene models and functional characterization of this region, we are not expecting changes in the chromosomal location of AKR1C genes or their protein coding potential.

In response to this comment and in order to underscore that AKR1C genes are functional and protein coding, we included in Table 2 information about the number of gene transcripts.

In addition, and as a response to Comment 5 by Reviewer 2, we expanded the text about the difficulties to accurately determine species orthologs for AKR1C genes in Discussion and included Supplementary Table S3 with a matrix of amino acid sequence percent identity between horse, human and mouse AKR1C putative orthologs.

Minor comments: Please provide more detailed information in the figure legends 2, 4, and 5.

Response: Legends for these three figures have been elaborated.

Reviewer 2 Report

In the manuscript a comprehensive molecular and cytogenetic characteristics of 200 kb deletion, reported earlier by other authors as copy number variation (CNV), was analyzed in a large cohort of horses. The applied methodology was appropriate and the obtained results are convincing.

Introduction:

Line 41. It is not true that cryptorchidism and hypospadias are minor forms of DSD. They are rather major forms of DSD.

Results - major concer:

It seems that some horses (Figure 4) could be classified as carrying 3 copies of the CNV, since in putative heterozygotes a relative number of copies was  approx.. 40, but there are horses with this value close to 120 or above 100. How to interpret it? Is it possible that a heterozygous horse will show 2 copies (2 copies on one homolog and deletion on the second homolog). It is suggest to reconsider accuracy of animal classification according to the detected deletion (N/N, N/D and D/D). May be genotypes should be classified according to number of copies at the studied CNVR (0, 1, 2, 3)?

Discussion:

More detailed information concerning metabolic effect of the lack of enzymes encoded by the deleted genes (Table 2) is necessary: Do steroid hormone levels in blood serum of horses homozygous or heterozygous for the deletion is affected? Does steroid hormone assay can be useful for a preliminary detection of D/D horses? Since the studied deletion is classified as CNV variant it would interesting to know how many variants at this CNVR were observed in horses - only 1 or 0? Were the number of copies at this CNVR analyzed with the use of other techniques (e.g. MLPA or ddPCR) in horses? Line 323. The following sentence is too general “Mouse models are less informative because there are no one-to-one mouse orthologs for human 323 AKR1C genes …”. It should be explain what level of similarity between mouse and human orthologs was estimated (for the coding sequences of the deleted genes and predicted amino acid sequences).

Author Response

Response to Reviewers

We thank the reviewers for the thorough, insightful, and constructive critique. As follows, please find our responses and list of revisions made in the manuscript. Please let us know if you would like us to provide any further information or clarification.

REVIEWER 2

Comment 1: Introduction, Line 41. It is not true that cryptorchidism and hypospadias are minor forms of DSD. They are rather major forms of DSD.

Response: Thank you, we removed ‘minor’ from line 55 in Introduction.

Comment 2: Results - major concern. It seems that some horses (Figure 4) could be classified as carrying 3 copies of the CNV, since in putative heterozygotes a relative number of copies was  approx.. 40, but there are horses with this value close to 120 or above 100. How to interpret it? Is it possible that a heterozygous horse will show 2 copies (2 copies on one homolog and deletion on the second homolog). It is suggest to reconsider accuracy of animal classification according to the detected deletion (N/N, N/D and D/D). May be genotypes should be classified according to number of copies at the studied CNVR (0, 1, 2, 3)?

Response: This is an excellent point and reflects the known multi-allelic nature of complex CNVs. This also means that both relative copy number quantitation by qPCR and absolute copy number quantitation by ddPCR will be of limited use for determining copy numbers on homologous chromosomes. The only exceptions are copy number = 0 in homozygous deletion and copy number = 1 in simple heterozygous deletion.

In order to reflect the possibility that this CNV can be multi-allelic, we reworded the text in Results 3.4. Detection of deletion heterozygotes by quantitative PCR and FISH on page 10, lines 311-321 as follows (:

“These results indicate that FISH analysis can clearly discriminate between horses with no deletion, deletion heterozygotes and horses with homozygous deletion. However, FISH is a method of too low throughput for routine screening of large number of horses. On the other hand, relative copy number estimation by qPCR to discriminate between horses with no deletion and those with heterozygous deletion remained ambiguous. Furthermore, a broad range in copy numbers as detected by qPCR analysis (Figure 4) suggests that this CNV may be multi-allelic like several common complex CNVs described in human diseases [40]. This also implies that qPCR-based analysis can confidently detect zero copies in homozygous deletion, maybe also 1 copy in simple heterozygotes, but will provide no information about the distribution of >2 copies on homologous chromosomes.”

We also edited Figure 4 by removing the red dotted horizontal line denoting putative threshold for heterozygotes.

Comment 3 Discussion: More detailed information concerning metabolic effect of the lack of enzymes encoded by the deleted genes (Table 2) is necessary: Do steroid hormone levels in blood serum of horses homozygous or heterozygous for the deletion is affected? Does steroid hormone assay can be useful for a preliminary detection of D/D horses?

Response: Indeed, information about hormone levels in relation to the deletion would be of great interest. It is just that we have very limited information about this. The majority of horses used in this study represent a 20-years collection of privately-owned horses submitted for cytogenetic analysis. In most cases, no additional clinical tests were conducted. Thus, in this study, we cannot match D/D genotypes with data on steroid hormone levels.

The good news is that 2 years ago, we started to freeze blood serum from all incoming cytogenetic cases. This will allow us to match hormone levels with cytogenetic findings and AKR1C region analysis in the future. We included a sentence about the need to combine genetic data with improved phenotype description (serum hormone analysis) in 5. Conclusions and Future Approaches as follows:

At the same time, for any further progress in the study of this deletion, it is important to improve phenotype descriptions for all new cases. For example, since the AKR1C genes are involved in steroid hormone biosynthesis, it would be important to include data on serum hormone levels.”

Comment 4: Since the studied deletion is classified as CNV variant it would interesting to know how many variants at this CNVR were observed in horses - only 1 or 0? Were the number of copies at this CNVR analyzed with the use of other techniques (e.g. MLPA or ddPCR) in horses?

Response: as discussed above in response to Comment 2, our qPCR results suggest that this CNV is multi-allelic, thus having several diffent copies on both homologs. As also discussed above, in cases with more than 2 copies, qPCR or ddPCR would not be able to tell whether these copies are on both homologs or on only one. Furthermore, with the sequence assembly in the region incomplete, design of ddPCR assay is complicated, though not impossible. Nevertheless, ddPCR assay bioinformatics design, optimization and testing on cases and controls is an elaborate process and needs a separate study.

Comment 5: Line 323. The following sentence is too general “Mouse models are less informative because there are no one-to-one mouse orthologs for human AKR1C genes …”. It should be explained what level of similarity between mouse and human orthologs was estimated (for the coding sequences of the deleted genes and predicted amino acid sequences).

Response: Thank you. In response we generated amino acid sequence identity matrix for all NCBI-annotated horse AKR1C genes, human AKR1C1-AKR1C4 and mouse Akr1c18 and included the data as Supplementary Table S3. We also elaborated Discussion page 13, lines 405-412 with regards sequence comparison between AKR1C paralogs and orthologs. The text reads as follows:

“Mouse models are less informative because there are no one-to-one mouse orthologs for human or horse AKR1C genes (Ensembl: http://www.ensembl.org/index.html). While amino acid sequence identity between human AKR1C1-AKR1C4 paralogs is 82-98% (Supplementary Table S3) [38], their one-to-one orthologs have been found only in primates, while orthologs in other mammals, including mouse, are many-to-many type (Ensembl: http://www.ensembl.org/index.html). One the one hand, this suggests rapid evolutionary divergence of the AKR1C orthologs across species, and on the other hand, difficulties to annotate paralogs within a species. The closest murine homolog for human AKR1C1-AKR1C4 genes is Akr1c18 (MGI: http://www.informatics.jax.org/marker) sharing only 65-68% amino acid sequence identity with human and 47-66% with equine orthologs (Supplementary Table S3).”

Round 2

Reviewer 1 Report

Authors have given prompt response to my earlier comments. However, the response does not satisfy this reviewer, and still my concern is not solved in the revised manuscript.

The starting sentence of section 3.3 is “The deleted region chr29:29,741,143-29,952,177 contains at least 2 protein coding genes by Ensembl and 6 protein coding and one RNA gene according to NCBI annotations (Table 2, Figure 3).

Although the horse chr.29 is not completely annotated, as the authors mentioned in the results and discussion section, my main concern is about NCBI annotation information shown in Fig. 3C and Table 2. They show 6 different NCBI IDs but represent only two genes (AKR1C3 and AKR1C23L), how this can be considered as 6 protein coding genes according to NCBI annotations. Moreover, please see the mRNA (XM_014737150) linked to LOC100070491. It is only partial, and does not contain starting codon (ATG) and initial amino acid (M).

At the other hand, Ensembl annotation clearly shows the above two genes in the deleted region. Moreover, the coverage area by ENSECAG00000021292 (AKR1C3) includes 5 or 6 of the NCBI IDs (also representing AKR1C3), and ENSECAG00000031302 (AKR1C23L) includes 1 NCBI ID (also representing AKR1C23L).

Therefore,

(1) please consider to exclude NCBI annotation information from Fig. 3C and table 2.

(2) write throughout the manuscript as “The deleted region contains at least 2 protein coding genes according to Ensembl horse genome annotation”.

(3) use the word “predicted” for mentioning AKR1C3 or AKR1C23L in horse, because their function is not yet characterized in horse.

(4) in table S3, keep AKR1C3_LOC100070501 and AKR1C23L_LOC100070528 (for horse), because these two are the sources for ensemble annotations.

Author Response

We thank the reviewer for thoroughness and patience in pointing out the weaknesses in how we have presented gene annotation in the deleted region. We agree that our first response to the critique was too prompt and definitely inadequate.

Indeed, the way the genes are presented in the manuscript is both confusing and wrong. Even though for most genomes the NCBI annotation is considered as the first source, in the case of the AKR1C-region the current annotation in Ensembl makes more sense. Furthermore, Genescan predictions in the UCSC Genome Browser for this region closely resemble Ensemble gene models, by presenting two unnamed coding genes with multiple mammalian homologs within the AKR1C gene family. Unfortunately, multi-tissue RNAseq data by equine FAANG project has been, as yet, aligned only to EquCab2 and is not adding much for this particular region.

In response we have revised the manuscript as follows:

  1. Text in chapter 3.3., page 7, lines 259-onwards:

The deleted region chr29:29,741,143-29,952,177 contains 2 predicted protein coding genes and one RNA gene by Ensembl (Figure 3). However, the Ensembl gene prediction ENSECAG00000021292 corresponds to 5 non-overlapping fragments of the same gene in NCBI (Table 2), suggesting that the region contains members of a gene family, which have not yet been accurately identified and annotated.”

  1. Removed NCBI predicted genes from Figure 3.
  2. Reworded Table 2 heading as:

Predicted genes in the deleted region by Ensembl and their counterparts by NCBI annotation

  1. Revised Table 2 which now clearly shows that the deleted region contains 2 predicted protein coding and one RNA gene by Ensembl and that the predicted AKR1C3 by Ensembl corresponds to five non-overlapping fragments of the same gene by NCBI.
  2. We talk now about two (2) protein coding genes in the deleted region throughout the manuscript (page 13, lines 388-390).
  3. The two genes are denoted now as "predicted”.
  4. Supplementary Table S3 contains now just 2 horse proteins - for loci LOC100070501 and LOC100070528. Minor changes are made in the text with regards amino acid sequence identity between species (page 14, lines 404 and 410).

Reviewer 2 Report

The manuscript was revised accordingly.

Author Response

Thank you!